# Predicting materials properties without crystal structure: deep representation learning from stoichiometry

Rhys E. A. Goodall [1] & Alpha A. Lee [1]✉

Machine learning has the potential to accelerate materials discovery by accurately predicting materials properties at a low computational cost. However, the model inputs remain a key stumbling block. Current methods typically use descriptors constructed from knowledge of either the full crystal structure — therefore only applicable to materials with already characterised structures — or structure-agnostic fixed-length representations hand-engineered from the stoichiometry. We develop a machine learning approach that takes only the stoichiometry as input and automatically learns appropriate and systematically improvable descriptors from data. Our key insight is to treat the stoichiometric formula as a dense weighted graph between elements. Compared to the state of the art for structure-agnostic methods, our approach achieves lower errors with less data.

[1] University of Cambridge, Cavendish Laboratory, Cambridge, UK. ✉email: aal44@cam.ac.uk

The discovery of new materials is key to making technologies cheaper, more functional, and more sustainable. However, the vastness of material space renders materials discovery via exhaustive experimentation infeasible. To address this shortcoming, significant effort has been directed towards calculating materials properties via high-throughput ab initio simulations[1–4]. However, ab initio simulations require atomic coordinates as input. These are typically only accessible for materials that have already been synthesised and characterised – only $O(10^5)$ crystal structures have been published[5], constituting a very limited region of the potential materials space[6]. A critical challenge exists for materials discovery in that expanding these ab initio efforts to look at novel compounds requires one to first predict the likely crystal structure for each compound. Ab-initio crystal structure prediction[7–9] is a computationally costly global optimisation problem which presents a significant challenge for high-throughput workflows. Whilst alternative strategies such as prototyping from known crystal structures[2,10] have been employed to manoeuvre around this bottleneck, identifying new stable compounds in a timely manner remains an important goal for computational material science.

One avenue that has shown promise for accelerating materials discovery workflows is materials informatics and machine learning. Here the aim is to use available experimental and ab initio data to construct accurate and computationally cheap statistical models that can be used to predict the properties of previously unseen materials and direct search efforts[11–13]. However, a key stumbling block to widespread application remains in defining suitable model inputs—so-called "descriptors". So far most applications of machine learning within material science have used descriptors based on knowledge of the crystal structure[14–20]. The use of structure-based descriptions means that the resulting models are therefore limited by the same structure bottlenecks as ab initio approaches when searching for novel compounds.

To circumvent the structure bottleneck, one approach is to develop descriptors from stoichiometry alone. In doing so we give up the ability to handle polymorphs for the ability to enumerate over a design space of novel compounds. This exchange empowers a new stage in materials discovery workflows where desirable and computationally cheap pre-processing models can be used, without knowledge of the crystal structure, to triage more time consuming and expensive calculations or experiments in a statistically principled manner.

Focusing on materials with a small and fixed number of elements, pioneering works[21–23] constructed descriptors by exhaustively searching through analytical expressions comprising combinations of atomic descriptors. However, the computational complexity of this approach scales exponentially with the number of constituting elements and is not applicable to materials with different numbers of elements or dopants. To address this shortcoming, general-purpose material descriptors, hand-curated from the weighted statistics of chosen atomic properties for the elements in a material, have been proposed[24–26]. However, the power of these general-purpose descriptors is circumscribed by the intuitions behind their construction.

In this paper, we develop a novel machine learning framework that learns the stoichiometry-to-descriptor map directly from data. Our key insight is to reformulate the stoichiometric formula of a material as a dense weighted graph between its elements. A message-passing neural network is then used to directly learn material descriptors. The advantage of this approach is that the descriptor becomes systematically improvable as more data becomes available. Our approach is inspired by breakthrough methods in chemistry that directly take a molecular graph as input and learn the optimal molecule-to-descriptor map from data[27,28].

We show that our model achieves lower errors and higher sample efficiency than commonly used models. Moreover, its learnt descriptors are transferable, allowing us to use data-abundant tasks to extract descriptors that can be used in data-poor tasks. We highlight the important role of uncertainty estimation to applications in material science and show how via the use of a *Deep Ensemble*[29] our model can produce useful uncertainty estimates.

## Results

**Representation learning of inorganic materials**. To eschew the hand engineering required by current structure-agnostic descriptor generation techniques, we represent each material's composition as a dense weighted graph. The nodes in this graph represent the different elements present in the composition and each node is weighted by the fractional abundance of the corresponding element. This novel representation for the stoichiometries of inorganic materials allows us to leverage neural message passing[28]. The message passing operations are used to update the representations of each of the element nodes such that they are contextually aware of the types and quantities of other elements present in the material. This process allows the model to learn material-specific representations for each of its constituent elements and pick up on physically relevant effects such as co-doping[30] that would otherwise be obscured within the construction of hand-engineered materials descriptors. We refer to this approach as *Roost* (**R**epresentati**o**n Learning fr**o**m **St**oichiometry). In the following paragraphs we introduce a specific model based on this idea.

To begin, each element in the model's input domain is represented by a vector. Whilst the only requirement is that each element has a unique vector, it can improve performance, particularly when training data is scarce, to embed elements into a vector space that captures some prior knowledge about correlations between elements[31,32]. These initial representations are then multiplied by a $n$ by $d - 1$ learnable weight matrix where $n$ is the size of the initial vector and $d$ is the size of the internal representations of elements used in the model. The final entry in the initial internal representation is the fractional weight of the element. A message-passing operation is then used to update these internal representations by propagating contextual information about the different elements present in the material between the nodes in the graph, Fig. 1 shows a schematic representation of this process. The mathematical form of the update process is

$$\boldsymbol{h}_i^{t+1} = U_t^{(h)}(\boldsymbol{h}_i^t, \boldsymbol{v}_i^t), \qquad (1)$$

where $\boldsymbol{h}_i^t$ is the feature vector for the $i^{th}$ element after $t$ updates, $\boldsymbol{v}_i^t = \{\boldsymbol{h}_\alpha^t, \boldsymbol{h}_\beta^t, \boldsymbol{h}_\gamma^t, ...\}$ is the set of other elements in the material's composition, and $U_t^{(h)}$ is the element update function for the $t + 1^{th}$ update. For this work, we use a weighted soft-attention mechanism for our element update functions. In general, attention mechanisms are used to tell models how important different features are for their given tasks. Soft-attention builds upon this concept by allowing the function that produces the attention coefficients to be learnt directly from the data. The soft-attention mechanism is the crux behind many state-of-the-art sequence-to-sequence models used in machine translation and language processing[33,34] and it has recently shown good results on graphs[35] and in some material science applications[36,37]. In this domain, the attention mechanism allows us to capture important materials concepts beyond the expressive power of older approaches e.g. that the properties and thus the representation

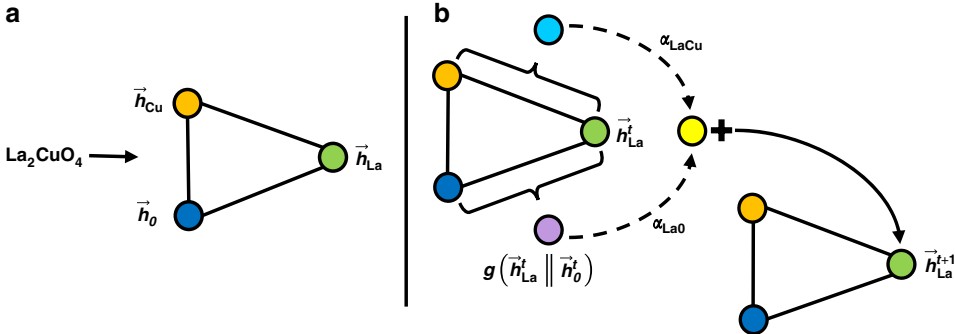

**Fig. 1 Schematic representation of stoichiometry graph and update rule. a** An example stoichiometry graph for La$_2$CuO$_4$. **b** A graphical representation of the the update function for the La representation. The pair dependent perturbations, shown as the cyan and purple nodes, are weighted according to their attention coefficients before being used to update the La representation.

of metallic atoms in a metal oxide should depend much more on the fact that oxygen is present than other metallic dopants being present.

The first stage of the attention mechanism is to compute unnormalised scalar coefficients, $e_{ij}$, across pairs of elements in the material,

$$e_{ij}^t = f^t(\boldsymbol{h}_i^t || \boldsymbol{h}_j^t), \qquad (2)$$

where $f^t(\ldots)$ is a single-hidden-layer neural network for the $t + 1^{th}$ update, the $j$ index runs over all the elements in $\boldsymbol{v}_i^t$, and $||$ is the concatenation operation. The coefficients $e_{ij}$ are directional depending on the concatenation order of $\boldsymbol{h}_i$ and $\boldsymbol{h}_j$. These coefficients are then normalised using a weighted softmax function where the weights, $w_j$, are the fractional weights of the elements in the composition,

$$a_{ij}^t = \frac{w_j \exp(e_{ij}^t)}{\sum_k w_k \exp(e_{ik}^t)}, \qquad (3)$$

where $j$ is a given element from $\boldsymbol{v}_i^t$ and the $k$ index runs over all the elements in $\boldsymbol{v}_i^t$. The elemental representations are then updated in a residual manner[38] with learnt pair-dependent perturbations weighted by these soft-attention coefficients,

$$\boldsymbol{h}_i^{t+1} = \boldsymbol{h}_i^t + \sum_{m,j} a_{ij}^{t,m} g^{t,m}(\boldsymbol{h}_i^t || \boldsymbol{h}_j^t), \qquad (4)$$

where $g^t(\ldots)$ is a single-hidden-layer neural network for the $t + 1^{th}$ update and the $j$ index again runs over all the elements in $\boldsymbol{v}_i^t$. We make use of multiple attention heads, indexed $m$, to stabilise the training and improve performance. The number of times the element update operation is repeated, $T$, as well as the number of attention heads, $M$, are hyperparameters of the model that must be set before training.

A fixed-length representation for each material is determined via another weighted soft-attention-based pooling operation that considers each element in the material in turn and decides, given its learnt representation, how much attention to pay to its presence when constructing the material's overall representation. Finally, these material representations are taken as the input to a feed-forward output neural network that makes target property predictions. Using neural networks for all the building blocks of the model ensures the whole model is end-to-end differentiable. This allows for its parameters to be trained via stochastic gradient-based optimisation methods. Whilst the rest of this paper focuses on regression tasks the model can be used for both regression and classification tasks by adapting the loss function and the architecture of the final output network as required.

**Uncertainty estimation**. A major strength of structure-agnostic models is that they can be used to screen large data sets of combinatorially generated candidates. However, most machine learning models are designed for interpolation tasks, thus predictions for materials that are out of the training distribution are often unreliable. During a combinatorial screening of novel compositions, we cannot assume that the distribution of new materials matches that of our training data. Therefore, in such applications, it becomes necessary to attempt to quantify the uncertainty of the predictions.

In statistical modelling there are two sources of uncertainty that are necessary to consider: First, the aleatoric uncertainty, which is the variability due to the natural randomness of the process (i.e. the measurement noise). Second, the epistemic uncertainty, which is related to the variance between the predictions of plausible models that could explain the data. This uncertainty arises due to having an insufficient or sparse sampling of the underlying process such that many distinct but equivalently good models exist for explaining the available data. Here we make use of a *Deep Ensemble* approach[29] that considers both forms of uncertainty.

Within a *Deep Ensemble* individual models require a proper scoring rule[39] to be used as the training criterion. To define a proper scoring rule for regression we consider the aleatoric uncertainty as part of a heteroskedastic problem formulation where the measurement noise depends on the position in the input space. The model is made to predict two outputs corresponding to the predictive mean, $\hat{\mu}_\theta(x_i)$, and the aleatoric variance, $\hat{\sigma}_{a,\theta}^2(x_i)$[40,41]. By assuming a probability distribution for the measurement noise we can obtain maximum likelihood estimates for the parameters of individual models by minimising a loss function proportional to the negative log-likelihood of the chosen distribution. Here we use a Laplace distribution which gives the loss function

$$\mathcal{L} = \sum_i \frac{\sqrt{2}}{\hat{\sigma}_{a,\theta}(x_i)} \| y_i - \hat{\mu}_\theta(x_i) \|_1 + \log(\hat{\sigma}_{a,\theta}(x_i)) \qquad (5)$$

Such loss functions are occasionally referred to as robust as they allow the model to learn to attenuate the importance of potentially anomalous training points.

To get an estimate for the epistemic uncertainty within the *Deep Ensemble* we generate a set of $W$ plausible sets of model parameters, $\{\hat{\theta}_1, ..., \hat{\theta}_W\}$, by training an ensemble of independent randomly-initialised models using the robust loss function (5). Due to the non-convex nature of the loss landscape, different initialisations typically end up in different local basins of attraction within the parameter space that have approximately equal losses[42]. We use these as samples of plausible sets of model

parameters to make Monte Carlo estimates for the expectation of the model, $\hat{y}(x_i)$, and the epistemic contribution to its variance, $\hat{\sigma}_e^2(x_i)$,

$$\hat{y}(x_i) = \int P(\hat{\theta}|x, y)\hat{\mu}_\theta(x_i)\, d\hat{\theta}$$
$$\simeq \frac{1}{W}\sum_w^W \hat{\mu}_{\theta_w}(x_i) \tag{6}$$

$$\hat{\sigma}_e^2(x_i) = \int P(\hat{\theta}|x, y)(\hat{y}(x_i) - \hat{\mu}_\theta(x_i))^2\, d\hat{\theta}$$
$$\simeq \frac{1}{W}\sum_w^W (\hat{y}(x_i) - \hat{\mu}_{\theta_w}(x_i))^2 \tag{7}$$

where $P(\hat{\theta}|x, y)$ is the hypothetical distribution of models that could explain the data. The effective marginalisation of $P(\hat{\theta}|x, y)$ from using an ensemble of models not only provides a way to estimate the epistemic uncertainty but also invariably leads to lower average errors. The total uncertainty of the ensemble expectation is simply the sum of the epistemic contribution and the average of the aleatoric contributions from each model in the ensemble.

$$\hat{\sigma}^2(x_i) = \hat{\sigma}_e^2(x_i) + \frac{1}{W}\sum_w^W \hat{\sigma}_{a,\theta_w}^2(x_i) \tag{8}$$

**Baseline model**. A common workhorse for the application of machine learning to both cheminformatics and materials science is Random Forests plus fixed-length descriptors[43,44].

Random Forests are a decision tree-based model that use an ensemble of multiple weak regressors known as trees[45]. Each of the trees is constructed to find a series of decision boundaries that split the data to minimise the squared deviations between the samples and the sample mean in each branch or leaf of the tree. Predictions are made by averaging over the outputs of the different trees when applied to new data. To overcome issues of over-fitting common to decision tree methods, Random Forests use bagging and random subspace projection to reduce the correlation between the trees improving their generalisation performance.

For our baseline inputs we use the general-purpose fixed-length *Magpie* feature vectors[24]. The *Magpie* feature set contains 145 features and is highly engineered to include as much prior scientific knowledge about the elements, stoichiometry, and electronic properties as possible.

**Data sets**. For this work, we consider a selection of experimental and ab initio data sets. The Open Quantum Materials Database (OQMD) data set contains the average formation enthalpy per atom calculated via density functional theory[1]. For comparison purposes we take the subset of 256,620 materials from[46], this subset contains only the lowest energy polymorph for each stoichiometry. The Materials Project (MP) data set we look at contains the band gaps for 43,921 non-metals present in the Materials Project catalogue[3]. As before we take only the lowest energy polymorph for each stoichiometry to ensure that the stoichiometry-to-property map is well defined. Finally, we consider a much smaller experimental data set consisting of 3895 non-metals for which the band gap has been measured experimentally (EX) as used in[25].

**Evaluation of sample efficiency**. Materials discovery workflows are often data limited. As a result, the sample efficiency of models is of critical importance. The sample efficiency can be investigated

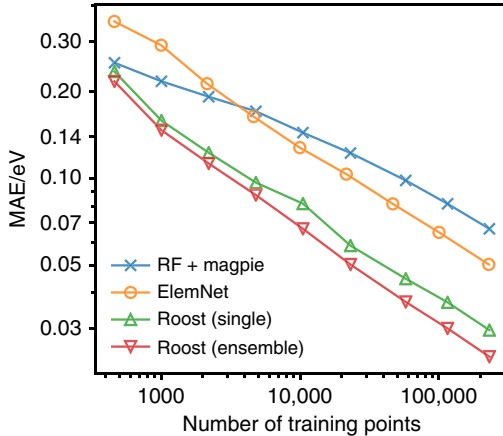

**Fig. 2 Sample efficiency learning curve on OQMD.** The figure shows learning curves for the OQMD data set as the amount of training data is varied for a fixed out-of-sample test set. Plotted on log-log scales the trends follow inverse power law as expected from statistical learning theory. Results for *ElemNet* taken from[46].

**Table 1 Performance Benchmarks on OQMD. The table shows the mean absolute error (MAE), and root mean squared error (RMSE) for the baseline and proposed models on 10% of the data that was randomly sampled and withheld as a test set. The bracketed numbers show the standard deviation in the last significant figure.**

|  | MAE/eV | RMSE/eV |
|---|---|---|
| RF + Magpie | 0.067 | 0.121 |
| ElemNet[46] | 0.055 | |
| Roost (Single) | 0.0297(7) | 0.0995(16) |
| Roost (Ensemble) | 0.0241 | 0.0871 |

by looking at how the performance of the model on a fixed test set changes as the model is exposed to more training data. From statistical learning theory, one can show that the average error for a model approximately follows an inverse power law relationship with the amount of training data in the large data limit[16,47]. As such the gradient and intercept on a log-log plot of the training set size against the model error indicate the sample efficiency of the model.

Figure 2 shows such learning curves for the OQMD data set and Table 1 records the benchmark results for when all the training data is used. In this case, 10% of the available data was held back from the training process as the test set. As well as our baseline model we also compare against *ElemNet*, an alternative neural network-based model that also takes the atomic fractions of each element as input[46]. The comparison shows that the inductive biases captured by the representation learning approach lead to a much higher sample efficiency. Indeed the crossover where *Roost* begins to outperform the traditional machine learning baseline occurs for $O(10^2)$ data points—a size typical of experimental databases collated for novel material classes[48,49] —as opposed to $O(10^3)$ for *ElemNet*.

**Evaluation of uncertainty estimates**. While the utility of stoichiometry-to-property models is primarily based on the amortisation of more time-consuming and expensive calculations or experiments, their approximate nature raises legitimate questions about when they can be used with confidence. Beyond simply building more sample-efficient models (e.g., by designing improved architectures or leveraging techniques such as transfer

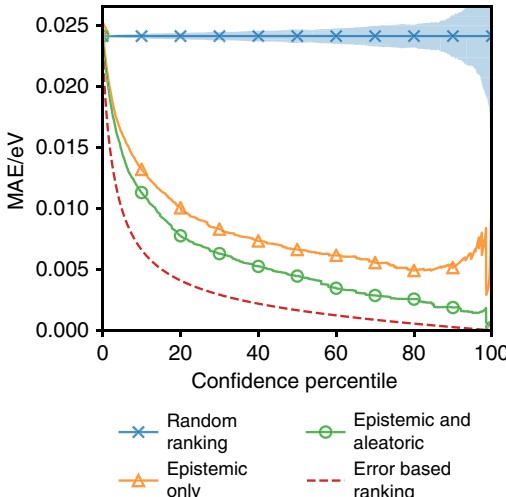

**Fig. 3 Confidence-error curves on OQMD.** The figure shows confidence-error curves on the OQMD test set. The curves show how the average model error changes as the data points the model is most uncertain about are removed sequentially. The random ranking-based curve in blue serves as a reference showing the result if all points are treated as having equal confidence, the blue shaded area highlights the curve's standard deviation computed over 500 random trials.

learning), well-behaved uncertainty estimates can allow for such models to be used with greater confidence (See Supplementary Note 1 for a discussion on the need for calibrated uncertainty estimates). Figure 3 highlights this idea on the OQMD data set. The plot shows how the test set error varies as a function of the confidence percentile. The error for a confidence percentile of $X$ is determined by re-calculating the average error of the model after removing the $X\%$ of the test set assigned the highest uncertainty by the model. Additional illustrative curves are included to show what would happen if the data was restricted in a random order and if the data was restricted according to the size of the model's error.

The added value of any form of uncertainty estimation is evident in large differences between the random ranking and the uncertainty-based curves—points with large uncertainties do on average have larger errors. On the other side, the error-based ranking curve provides a useful lower bound for comparison about how good those uncertainty estimates are. However, it should be noted that optimal uncertainties would not result in exact coincidence with this error-based ranking curve. This is due to instances where the model might make accurate predictions despite those predictions not being well supported by the training data, in which case the model should have high uncertainty. These points would be removed early in any uncertainty-based curve but late in the error-based ranking curve resulting in the uncertainty-based curve being higher than the error-based ranking curve.

To highlight the benefit of using a full framework for estimating the uncertainty, one that considers both aleatoric and epistemic uncertainties, we compare against a purely epistemic alternative based on an ensemble of similar models that only estimate a predictive mean and are trained using an L1 loss function. We see that whilst the two ensembles have comparable errors over the whole data set, the full framework gives more reliable uncertainty estimates shown by the curve for the full framework (Epistemic & Aleatoric) decreasing more steeply than the curve for the epistemic-only alternative. Within the full framework the relative

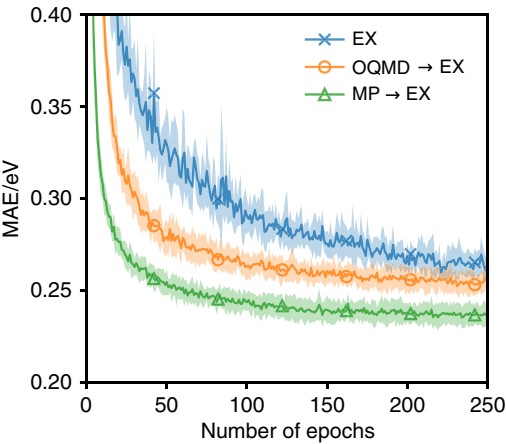

**Fig. 4 Test set error during training on EX.** The figure shows how the MAE on the test set changes throughout the training of the model for different transfer learning scenarios. The curves show the average MAE over 10 independent randomly-initialised models with the shaded area corresponding to the standard deviation of the models at each point.

magnitudes for the epistemic and aleatoric components vary depending on the data set being investigated and the extent to which the model is being tested in an interpolative regime (see Supplementary Figs. 4 and 5). This implies that the different forms of uncertainty capture different effects in the data and further supports the use of a full framework.

**Transfer learning.** For experimental data sets with smaller numbers of data points traditional machine learning methods based on decision tree or kernel models have historically tended to perform comparably if not better than deep neural network-based models. However, a strength of neural network-based models over such methods is that they are much more amenable to transfer learning[50]. Transfer learning focuses on using knowledge gained from one problem to achieve faster optimisation and/or a lower error on another problem.

As a result of substantial efforts, data-sets derived via high-throughput ab initio workflows can be many times larger than their experimental cousins, making them ripe for transfer learning[51]. To investigate the extent to which transfer learning helps our model we train three sets of models on the EX data set. The first set is directly trained on EX, the second is first trained on OQMD then fine-tuned on EX (OQMD → EX), and the third is trained on MP before fine-tuning on EX (MP → EX). Due to the similarity of the MP and EX tasks, to ensure any changes in performance observed are not artefacts of the experimental design, we remove all materials from the MP data set that are also found in the EX data set such that the two are independent. For all these experiments the same 20% of EX was withheld as an independent test set.

A benefit of learning material descriptors is that similarity between the descriptors of different materials learnt for a given task should be relevant for other tasks, therefore allowing non-cognate transfer learning. We see this in Fig. 4 where transfer learning from OQMD leads to faster convergence and slightly lower errors on the EX data set than direct training despite the mismatch between the tasks. If the tasks are cognate, as is the case between MP and EX, the benefits of transfer learning are even more pronounced. Here, in addition to the benefits of having pre-trained the message passing sections of the model, the pre-trained weights of the output network give a strong inductive bias for

**Table 2 Transfer Learning Benchmarking on EX.** The table shows the ensemble mean absolute error (MAE), and root mean squared error (RMSE) for the three transfer learning scenarios and baselines on 20% of the data that was randomly sampled and withheld as a test set.

|  | MAE/eV | RMSE/eV |
|---|---|---|
| Baseline EX | 0.277 | 0.460 |
| SVM EX[25] |  | 0.45 |
| Roost EX | 0.243 | 0.422 |
| Roost OQMD → EX | 0.240 | 0.404 |
| Roost MP → EX | 0.219 | 0.364 |

fitting the materials descriptor-to-property mapping resulting in notably lower predictive errors (Table 2).

**Ablation study**. The proposed reference model incorporates many different ideas to build upon previous work in the materials informatics and machine learning communities. Therefore, we have conducted an ablation study to show which parts of the model are most important for its enhanced performance. We examined the following design choices:

1. The use of an element embedding that captures correlations between elements versus a OneHot embedding of elements,
2. The use of a robust loss function based on the negative log-likelihood of a Laplace distribution (5) against the use of a standard L1 loss function,
3. Whether it is best to include the fractional element weights as node-level features, within the pooling operation, or in both places,
4. The use of our weighted soft-attention-based pooling throughout the architecture versus an alternative mean-based pooling mechanism,
5. The use of residual architectures for both the message passing and output neural networks, and
6. The impact on model performance from only using the message passing section of the model without an output network.

The combinations of design choices examined are shown in Table 3. We train 10 randomly-initialised models for each design choice. We look at both the statistics across these single models to allow for the significance of different choices to be understood as well as their ensembled performance. We repeat the ablation study for both the EX and OQMD data sets to allow us to understand how different design choices trade-off in the small and large data limits. The results are shown in Table 4.

The primary conclusion from the ablation study is that whilst the design choices made in the reference architecture described do lead to slight improvements in performance, all models from the ablation study (with exception of Model 3 that does not include the element weights) still significantly out-perform alternative models such as *ElemNet* or the Random Forest plus *Magpie* baseline on the OQMD data set. As such, it is apparent that it is the Roost framework's approach of reformulating the problem as one of regression over a multiset and not specific architectural details that is responsible for the observed improvements.

Comparing the reference model and Model 1 we see that the choice of an element embedding that captures chemical correlation leads to improved model performance on the smaller EX data set but does not result in significant differences for the larger OQMD data set. This suggests that the models can learn to compensate for the lack of domain knowledge if sufficiently large

amounts of data are available. This result supports our claim that end-to-end featurization continuously improves as the model is exposed to more data.

The robust loss function (5) performs comparably on the EX data set to a more conventional L1 loss function (Model 2). Given that they offer similar average errors the use of a robust loss function is highly compelling even for single models as it also provides an estimate of the aleatoric uncertainty with minimal computational overhead. Looking at the OQMD data set the distinction between the two different loss functions is more apparent. The attenuation effect of the robust loss, that it can suppress the need to fit outliers, is observed in how the reference model achieves a lower MAE but a higher RMSE than Model 2. When proceeding to ensemble the single models, the validity of such a mechanism becomes apparent as both the MAE and RMSE are lower for the reference model in the ensembled case. This can be attributed to the cancellation of errors amongst predictions on the outlying (high squared-error) data points when ensembling.

Models 3, 4 and 5 from the ablation study look at how including the fractional element weights in different ways influences the model performance. As expected we see that omitting the element weights entirely in Model 3 leads to an order of magnitude decrease in performance on the OQMD data set. However, whilst there is still a significant decrease in performance for the EX data set the error is still relatively comparable to that achieved by the standard model. This is due to a lack of diversity within different chemical sub-spaces in the EX data set. As a consequence, the EX data set is perhaps a less discriminative benchmark than OQMD despite the challenges associated with data scarcity. Including the weights on both the nodes and via the pooling operation gave the best results being marginally better than solely including the element weights on the nodes. Only including the weights via the pooling operation gave slightly worse results. This can be explained from the relative lack of information as the weighted soft-attention-based pooling (3) only includes the weights of the second element in the pairing as opposed to both elements if the weights are included as node features.

Whilst we primarily make use of a soft-attention-based pooling mechanism alternative pooling mechanisms are feasible. In Model 6 we replace the pooling operations with a mean-pooling mechanism of the form

$$\boldsymbol{h}_i^{t+1} = \boldsymbol{h}_i^t + \frac{1}{J}\sum_{j=1}^{J} g^t(\boldsymbol{h}_i^t || \boldsymbol{h}_j^t), \qquad (9)$$

where $\boldsymbol{h}_i^t$ is the internal representation of the $i^{th}$ element after $t$ updates, $g^t(...)$ is a single-hidden-layer neural network for the $t+1^{th}$ update, $||$ is the concatenation operation and the $j$ index runs from 1 to $J$ over the set $\boldsymbol{v}_i^t$ which contains the other elements in the material's composition. This model achieves a lower RMSE but has a higher MAE when considering individual models. However, when the models are ensembled the soft-attention-based pooling mechanism achieves both lower MAE and RMSE. This suggests that there is scope to tailor the reference model presented here for different applications by conducting neural architecture searches. However, this is an extremely computationally expensive process beyond the scope of this work[52].

Comparing Models 7, 8, and 9 we see that using residual architectures in both the message passing stages and the output network lead to improved performance. Interestingly we see that replacing the output network with a single linear transformation (Model 10) does not significantly impact the performance of single models on the OQMD data set but does result in worse performance from the ensembled models. A potential explanation for this comes from considering the effective prior of the model

**Table 3 Model Design Choices for Ablation Study. The table shows the different model architectures based on the Roost framework studied in the ablation study.**

| | Reference | Model 1 | Model 2 | Model 3 | Model 4 | Model 5 | Model 6 | Model 7 | Model 8 | Model 9 | Model 10 |
|---|---|---|---|---|---|---|---|---|---|---|---|
| Matscholar Element Embedding[32] | ✓ | | ✓ | ✓ | ✓ | ✓ | ✓ | ✓ | ✓ | ✓ | ✓ |
| OneHot Element Embedding | | ✓ | | | | | | | | | |
| Robust Loss Function | ✓ | ✓ | | ✓ | ✓ | ✓ | ✓ | ✓ | ✓ | ✓ | ✓ |
| Weights on Nodes | ✓ | ✓ | ✓ | | ✓ | ✓ | ✓ | ✓ | ✓ | ✓ | ✓ |
| Weights in Pooling | ✓ | ✓ | ✓ | ✓ | ✓ | ✓ | ✓ | ✓ | ✓ | ✓ | ✓ |
| Soft Attention Pooling | ✓ | ✓ | ✓ | ✓ | ✓ | ✓ | | ✓ | ✓ | ✓ | ✓ |
| Mean Pooling | | | | | | | ✓ | | | | |
| Residuals when Message Passing | ✓ | ✓ | ✓ | ✓ | ✓ | ✓ | ✓ | | ✓ | ✓ | ✓ |
| Residuals in Output Network | ✓ | ✓ | ✓ | ✓ | ✓ | ✓ | ✓ | ✓ | | ✓ | ✓ |
| No Output Network | | | | | | | | | | | ✓ |

**Table 4 Ablation Study Model Performances. The table shows the how the performance varies for different model architectures based on the Roost framework. Numbers in parentheses are used to show the standard error in the last significant figure. The lowest values in each row are highlighted in bold.**

| | | Reference | Model 1 | Model 2 | Model 3 | Model 4 | Model 5 | Model 6 | Model 7 | Model 8 | Model 9 | Model 10 |
|---|---|---|---|---|---|---|---|---|---|---|---|---|
| EX | MAE | **0.264(3)** | 0.279(2) | 0.269(2) | 0.329(2) | 0.274(3) | 0.269(3) | 0.269(2) | 0.284(3) | 0.271(2) | 0.292(3) | 0.273(2) |
| | RMSE | **0.448(4)** | 0.476(4) | **0.447(4)** | 0.529(4) | 0.476(6) | 0.454(4) | 0.459(5) | 0.477(7) | 0.465(5) | 0.490(4) | 0.470(5) |
| OQMD | MAE | **0.0297(2)** | **0.0295(1)** | 0.0310(2) | 0.1673(2) | 0.0317(2) | 0.0300(2) | 0.0313(3) | 0.0320(3) | 0.0306(4) | 0.0330(3) | 0.0299(1) |
| | RMSE | 0.0995(5) | 0.0992(3) | 0.0979(4) | 0.2710(3) | 0.1022(6) | 0.0999(5) | **0.0959(6)** | 0.1025(6) | 0.1017(9) | 0.1040(6) | 0.0981(4) |
| EX ens | MAE | **0.243** | 0.260 | 0.249 | 0.312 | 0.251 | 0.251 | 0.250 | 0.259 | 0.251 | 0.265 | 0.255 |
| | RMSE | **0.422** | 0.445 | 0.423 | 0.501 | 0.450 | 0.428 | 0.435 | 0.442 | 0.435 | 0.451 | 0.444 |
| OQMD ens | MAE | **0.0241** | **0.0241** | 0.0248 | 0.1644 | 0.0256 | 0.0243 | 0.0248 | 0.0253 | 0.0247 | 0.0259 | 0.0249 |
| | RMSE | **0.0871** | 0.0878 | 0.0882 | 0.2682 | 0.0898 | 0.0874 | 0.0875 | 0.0880 | 0.0877 | 0.0885 | 0.0911 |

without an output network. The addition of the output network changes the form of the prior hypothesis space of the model and as a result the distribution of distinct local basins of attraction[53]. The reduced benefits of model averaging within the ensemble for models without output networks could potentially be due to changes in the loss landscape meaning that such models are more likely to end up in correlated basins of attraction.

## Discussion

We propose a novel and physically motivated machine learning framework for tackling the problem of predicting materials properties without crystal structures. Our key methodological insight is to represent the compositions of materials as dense weighted graphs. We show that this formulation significantly improves the sample efficiency of the model compared to other structure-agnostic approaches.

Through modelling both the uncertainty in the physical process and in our modelling processes, the model produces useful estimates of its own uncertainty. We demonstrate this by showing that as we restrict, according to our uncertainty estimates, the confidence percentile under consideration, we observe steady decreases in the average error on the test set. Such behaviour is important if we wish to use our model to drive an activate learning cycle.

We show that the representations learnt by the model are transferable allowing us to leverage data-abundant databases, such as those obtained by high-throughput ab initio workflows, to improve model performance when investigating smaller experimental data sets. The ability of the model to transfer its learnt descriptors suggests that self-supervised learning may be a viable avenue to bolster model performance[54,55].

We have conducted an extensive ablation study to examine the model. We show that it is the reformulation of the problem such that both the descriptor and the fit are learnt simultaneously that results in the improved performance, not the specific details of the message passing architecture used.

More broadly, the *Roost* framework's ability to handle multi-sets of various sizes makes it applicable to other important problems in material science such as the prediction of the major products of inorganic reactions[56]. We believe that recasting more problems in material science into this language of set regression, using the same message passing framework as our *Roost* approach or other frameworks[57,58], provides an exciting new area for the development of novel machine learning methods.

## Methods

In this work, we adopt the same architecture and hyperparameters for all the problems investigated. These choices were made based on heuristic ideas from other graph convolution-based architectures.

We use the *Matscholar* embedding from[32] for which $n = 200$. We chose an internal representation size of $d = 64$ based on the *CGCNN* model[18].

We opted to use 3 message passing layers based on the default configuration of the *MEGNet* model[19]. For the choice of neural networks to use within our weighted soft-attention-based pooling function we drew inspiration from the *GAT* architectures presented in[35] which led to us choosing single-hidden-layer neural networks with 256 hidden units and LeakyReLU activation functions for $f^t(...)$ and $g^t(...)$. For the reference model, we used 3 attention heads in each of message passing layers.

The output network used for the reference model is a deep neural network with 5 hidden layers and ReLU activation functions. The number of hidden units in each layer is 1024, 512, 256, 126, and 64 respectively. Skip connections were added to the output network to help tackle the vanishing gradient problem[38].

The sizes of various networks were selected to ensure that our model was appropriately over-parameterised for the OQMD data set. For modern neural network architectures over-parameterisation leads to improved model performance[59,60]. Our reference model has 2.4 million parameters – approximately 10x the size of the OQMD training set used.

For numerical reasons when estimating the aleatoric uncertainty the model is made to predict $\log(\hat{\sigma}_a(x_i))$ which is then exponentiated to get $\hat{\sigma}_a(x_i)$. In this work

we use ensembles of $W = 10$ to estimate the epistemic contribution within the *Deep Ensemble*.

All transfer learning experiments were conducted as warm-restarts with all of the model parameters being re-optimised given the new data. This was observed to give better performance than freezing the message passing layers and only re-optimising the weights of the output neural network.

The mean-based pooling function in the ablation study used single-hidden-layer neural networks with 256 hidden units and LeakyReLU activation functions for $g^t$ ( . . .).

All the neural network-based models examined in both the main results and the ablation study were trained using the Adam optimiser and fixed learning rate of $3 \times 10^{-4}$. A mini-batch size of 128 and weight decay parameter of $10^{-6}$ were used for all the experiments. The models were trained for 250 epochs (cycles through the training set).

For our baseline models we use the Random Forest implementation from *scikit-learn* and use *Matminer*[61] to generate the *Magpie* features. The max features and number of estimators for the Random Forest are set to 0.25 and 200 respectively.

## Data availability

The OQMD data set used for this work was collated from the openly available Open Quantum Materials Database at http://oqmd.org[1,2]. We use the subset of OQMD studied in ref. [46]. The MP data set used for this work was collated using the Materials API[62] from the openly available Materials Project database at https://materialsproject.org[3]. The EX data set used is available alongside[31]. Exact copies of the data sets studied and the results needed to generate the figures presented in the manuscript are released alongside the source code at https://doi.org/10.5281/zenodo.4133793.

## Code availability

An open-source static implementation of the model, including a list of experiments required to replicate the main results of this work, is available from https://doi.org/10.5281/zenodo.4133793.

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

## Acknowledgements

The authors acknowledge the support of the Winton Programme for the Physics of Sustainability. A.A.L. acknowledges the funding from the Engineering and Physical Sciences Research Council (EPSRC)—EP/S003053/1. The authors also thank Janosh Riebesell for their feedback and suggestions.

## Author contributions

R.E.A.G. and A.A.L. devised the idea for the project. R.E.A.G. implemented the idea and conducted the experiments. R.E.A.G. and A.A.L. interpreted the results and prepared the manuscript.

## Competing interests

The authors declare no competing interests.
