## [Peer Review File · Nature Communications]

REVIEWER COMMENTS

Reviewer #1 (Remarks to the Author):

The authors present a manuscript describing the construction of a new machine learning algorithm architecture, Roost. This new approach is novel and potentially very important. The authors argue that the attention-based mechanism which utilizes a graph representation of the stoichiometry outperforms state of the art descriptors including those, like Magpie, which include chemical domain knowledge. This is an important advance in the field especially given the shortcomings of structure-based ML like CGCNN and others. I think this paper should absolutely be published and I believe that the readership of Nature Communications would find it of great value.

I do have a key lingering doubt for the authors to consider prior to publication. It would be prudent to first perform an ablation study to ensure that the key enhancement is due to the novel advances put forth in this work. Specifically, I suggest the authors remove the attention section of the code and explore how this affects performance.

Reviewer #2 (Remarks to the Author):

This paper describes a new method for the prediction of the properties of inorganic materials from their elemental compositions. The method, named "Roost," is rather clever and contains a few different advancements that each may lead to its improved accuracy over state-of-the-art methods. The advancements presented here are of the impact needed for publication in Nature Communications, but the quality of the study of which innovations leads the improved score is poor. For that reason, I recommend this paper to published in Nature Communications after major revisions.

Roost presents several interesting advancements over ElemNet, a previous NN applied to the prediction of material properties and other NN in this general field (e.g., SchNet):

1. Seeding the initial representation for elements from known material properties
2. Applying a random matrix to create initial features as linear combinations of these properties
3. Using a message passing architecture to compute updated representations for the elements
4. Condensing the per-element representations of the material to a whole-composition representation using soft attention

The above advances are thought-provoking and the fact that they outperform existing strategies will certainly attract research that builds off these innovations. My concern is that the paper does not identify which is the key ingredient, if any single innovation, leading to the improved performance. I am left with questions such as:

Does pre-seeding the representation already-trained elemental features lead to the improvement over ElemNet, which trains all weights from initial randomization?

Are the message passing layers more effective at learning improved representations than adding additional fully-connected layers?

Is using a soft-attention layer to combine elemental features during the message-passing or readout (i.e., going from node-level to graph-level features) more effective than performing a weighted sum of each element based on elemental fractions? Having the weights (varying between 0-1) multiplied by an exponential of another scalar (which could easily vary into the 100s) also

makes me curious about whether the elemental fraction has any effect at all.

I am generally curious to figure out what is it about this structure that is leading to the better performance. My concern is the paper needs to achieve a higher standard evaluating how the design of the network influences performance for higher-impact publication.

Some minor comments of places the paper requires further clarity:

- How do you decide the variance of the input features when computing the aleatoric uncertainty (e.g., $\sigma_{a,i}$ in Eq. 5)? Are those the parameters being determined with maximum likelihood estimation?

- How is Fig. 4 computed? Is the MAE at 40%, for example, the MAE of the top 60% of training entries based on your metrics? Please clarify.

Logan Ward

Reviewer #3 (Remarks to the Author):

This work shows a novel machine learning approach by applying graph neural network ideas to stoichiometry. The approach is empirically validated on 3 data sets (2 computational, 1 experimental) and the use of transfer learning is explored.

One of the novel additions on the machine learning side is the way uncertainty is handled, though this in particular requires some more analysis (see details below).

Outside of this issue, the work is well constructed, using a novel but reasonable machine learning approach with appropriate technical evaluation. This will be of significant interest to researchers already doing machine learning for materials systems. However, I do not think this work will be of particularly broad scientific interest and I believe it is more appropriate to a more narrow journal.

Before publication in any journal, I have just two significant concerns and a number of smaller issues focussed on the clarity of presentation, some of which affect the technical correctness and precision of what is being said..

Major issues

1. The attempt to separate out aleatoric and epistemic uncertainty is unusual. I was not aware of the Kendall and Gal (NIPS 2017) work, but reviewing that, there are significant differences in how your model is attempting to capture this (notably their use of a Bayesian Neural Network). Therefore, I think you have a higher standard to show that this is doing the right thing. I do not think the evaluation culminating in Figure 4 accomplishes this. The questions that first come to mind that are left unresolved are:

* What is the coverage probability? The figure shows ranking, but that's not what the uncertainty modelled attempted to capture. If the model says the standard deviation of the error is X, is it actually X?

* What's the relative contributions / magnitudes of the two error components?

* Is using one of the other error models insufficient?

2. The methods section does not describe a hyperparameter search procedure. Depending on the procedure used, it could raise questions about the validity of the results on the test set. Related, while the Roost code is open sourced, I looked through the repository and did not see a direct way to reproduce the models described in this paper. I would expect code that directly reproduces the models to be included in some way either in the Roost repository directly or as supplemental files with this publication.

Minor issues

1. The end of the first paragraph in the introduction: This is not actually fair to how high-

throughput workflows actually work. A number of simplifications (like assuming atomic substitutions produce approximately the same crystal structure) are used to avoid the costly global optimization. So it's not a "prohibitive bottleneck", it's something that is worked around.

2. s/statistically principle manner/statistically principled manner/

3. s/n is the initial vector/n is the size of the initial vector/

4. Below equation (2): You refer to $f^t(\dots)$. I suspect this is a typo and should be $f(\dots)$. Unless you are implying a step dependent f function (which is a fine architecture choice), but you need to be clear, including in equation (2)

5. Associated with equation (2): You really should be clear that these edge terms are directional. e_{ij} is used to update node i . I had to work through the next several equations before that was clear.

6. Eq (3): Your notation is sloppy, you have overloaded j . The denominator sum should use a different index.

7. Eq (4): It's worth noting in the text that you use a residual net structure. This is a clear design choice that not all graph nets use.

8. Eq (5) and Eq (6). You never define F or y_i . This whole part would probably be simpler to use F in equation (5) and avoid $y^{\hat{i}}$

9. Eq (6). $\theta^{\hat{m}}$ should be $\theta^{\hat{w}}$

10. Fig 3: The shading is not defined.

11. Fig 3: I can't make sense out of the caption. You say it is "error on the test set" but then say "training curve" which is at least ambiguous whether it's training or test set error.

12. The methods section does not contain any details for the transfer learning. In particular, how you either limited fine tuning training (or not) or used frozen layers (or not).

REVIEWER COMMENTS

Reviewer #1 (Remarks to the Author):

The authors present a manuscript describing the construction of a new machine-learning algorithm architecture, Roost. This new approach is novel and potentially very important. The authors argue that the attention-based mechanism which utilizes a graph representation of the stoichiometry outperforms state of the art descriptors including those, like Magpie, which include chemical domain knowledge. This is an important advance in the field especially given the shortcomings of structure-based ML like CGCNN and others. I think this paper should absolutely be published and I believe that the readership of Nature Communications would find it of great value.

We are glad that the referee agrees that the approach we introduce is an important advance suitable for publication in Nature Communications.

I do have a key lingering doubt for the authors to consider prior to publication. It would be prudent to first perform an ablation study to ensure that the key enhancement is due to the novel advances put forth in this work. Specifically, I suggest the authors remove the attention section of the code and explore how this affects performance.

In the revised manuscript we have included an ablation study exploring different design decisions made in the construction of our model. We hope that this will allow readers to better understand which aspects are important.

Concretely, Model 6 from the ablation study replaces the soft-attention based pooling function with an alternative mean based pooling function as suggested. Whilst this does lead to a marginal decrease in model performance Model 6 still significantly outperforms alternative neural network based architectures such as ElemNet as well as the baseline Random Forest model we compare against. This result shows that it is the novel representation of stoichiometry as a dense weighted graph and the propagation of contextual information on this graph to produce contextually aware material descriptors that is the key enhancement. The wider ablation study shows that this enhancement is robust to adaptations to the architecture described in the manuscript.

Reviewer #2 (Remarks to the Author):

This paper describes a new method for the prediction of the properties of inorganic materials from their elemental compositions. The method, named "Roost," is rather clever and contains a few different advancements that each may lead to its improved accuracy over state-of-the-art methods. The advancements presented here are of the impact needed for publication in Nature Communications, but the quality of the study of which innovations lead to the improved score is poor. For that reason, I recommend this paper to be published in Nature Communications after major revisions.

We are glad that the referee agrees that the contributions set out in our work are important and suitable for publication in Nature Communications.

Roost presents several interesting advancements over ElemNet, a previous NN applied to the prediction of material properties and other NN in this general field (e.g., SchNet):

- 1. Seeding the initial representation for elements from known material properties*
- 2. Applying a random matrix to create initial features as linear combinations of these properties*
- 3. Using a message passing architecture to compute updated representations for the elements*
- 4. Condensing the per-element representations of the material to a whole-composition representation using soft attention*

The above advances are thought-provoking and the fact that they outperform existing strategies will certainly attract research that builds off these innovations. My concern is that the paper does not identify which is the key ingredient, if any single innovation, leading to the improved performance.

In the revised manuscript, we have performed a comprehensive ablation study to isolate which parts of the model lead to the improved performance. We believe that this makes it much clearer for readers which contributions are most significant.

I am left with questions such as:

Does pre-seeding the representation with already-trained elemental features lead to the improvement over ElemNet, which trains all weights from initial randomization?

This is dependent on the size of the dataset. Comparing the reference model with Model 1 from the ablation study, pre-seeding the representation with already-trained elemental features is observed to improve the model performance for the smaller EX data set compared to using an OneHot embedding. However, for the larger OQMD data set there is no tangible benefit from using an initial representation that captures chemical correlations.

Are the message passing layers more effective at learning improved representations than adding additional fully-connected layers?

Model 10 in the ablation study explores this idea. In this experiment, we replace the output network with a linear transformation, and observe that this model performs similarly to the reference model. This shows that the removal of additional fully-connected layers does not significantly harm the model's ability to learn. Which suggests that the propagation of contextual information between distinct elements using message passing layers before pooling into a fixed length vector is more effective than trying to learn the same patterns using fully-connected layers operating on a fixed length vector.

This observation does not exclude the possibility that for large data sets, such as OQMD, there might be additional benefits from adding more layers (either fully connected or message passing) but doing so would be task dependent and could potentially harm the performance on small data sets like EX.

Is using a soft-attention layer to combine elemental features during the message-passing or readout (i.e., going from node-level to graph-level features) more effective than performing a weighted sum of each element based on elemental fractions? Having the weights (varying between 0-1) multiplied by an exponential of another scaler (which could easily vary into the 100s) also makes me curious about whether the elemental fraction has any effect at all.

In the ablation study we show that including the weights as a node level feature alone gives a better result than only including the weights via the attention mechanism. However, only including the weights via the attention mechanism still performs much better than non-Roost models on OQMD. We observed that including the weights in both places led to marginally better performance for both EX and OQMD. As expected not including the weights in any manner considerably harms the model performance.

In Model 6 from the ablation study we replace the soft-attention based pooling function with an alternative mean based pooling function for both for the message-passing and the readout pooling operations. Here the element weights were included as node level features. This change led to only a small decrease in model performance showing that, whilst helpful, the use of soft-attention based pooling function is not critical to the improved performance.

I am generally curious to figure out what it is about this structure that is leading to the better performance. My concern is the paper needs to achieve a higher standard evaluating how the design of the network influences performance for higher-impact publication.

The ablation study included in the revised manuscript shows that it is the novel representation of stoichiometry as a dense weighted graph and the propagation of contextual information on this graph to produce contextually aware material descriptors that is the key enhancement. This enhancement is robust to adaptations to the architecture described in the manuscript.

Some minor comments of places the paper requires further clarity:

- How do you decide the variance of the input features when computing the aleatoric uncertainty (e.g., $\sigma_{a,i}$ in Eq. 5)? Are those the parameters being determined with maximum likelihood estimation?

We have re-written this section and amended the notation used to make this clearer in the revised manuscript.

In the robust loss function both the target label and the aleatoric contribution to the uncertainty are learned via maximum likelihood estimation. To do this we have to assume a form for the measurement noise distribution of the target value and then minimise the negative log likelihood. We do not make any assumptions about how the input features are distributed nor about how the model parameters are distributed.

We show the benefit of using a robust loss function in Fig. 4 (now Fig. 3) which shows that we get much better uncertainty estimates from ensembling models trained with such a robust loss (green curve) as opposed to an L1 loss (orange curve).

- How is Fig. 4 computed? Is the MAE at 40%, for example, the MAE of the top 60% of training entries based on your metrics? Please clarify.

In the revised manuscript, we have amended the text to clarify how Fig. 4 (now Fig. 3) was produced.

The plot shows how the test set error varies as a function of confidence percentile – The error for a confidence percentile of X is determined by re-calculating the average error of the model after removing the X% of the test set assigned the highest uncertainty by the model.

As suggested this means that the error at a confidence percentile of 40% is the error on the remaining 60% of the data after the 40% of the data about which the model is least confident about (or most uncertain about) has been removed.

Reviewer #3 (Remarks to the Author):

This work shows a novel machine learning approach by applying graph neural network ideas to stoichiometry. The approach is empirically validated on 3 data sets (2 computational, 1 experimental) and the use of transfer learning is explored. One of the novel additions on the machine learning side is the way uncertainty is handled, though this, in particular, requires some more analysis (see details below).

Outside of this issue, the work is well constructed, using a novel but reasonable machine learning approach with appropriate technical evaluation. This will be of significant interest to researchers already doing machine learning for materials systems. However, I do not think this work will be of particularly broad scientific interest and I believe it is more appropriate to a more narrow journal.

We are glad that the referee agrees that our work is well constructed and will be of significant interest to many researchers working in the emerging field of machine learning for materials systems.

We also believe that the work will also be of wider interest for the readers of Nature Communications due to the fact that it contributes to a limited literature on machine learning for sets and multisets and provides a tangible and important use case for such work by highlighting that such problems exist in material science.

Before publication in any journal, I have just two significant concerns and a number of smaller issues focussed on the clarity of presentation, some of which affect the technical correctness and precision of what is being said.

Major issues

- 1. The attempt to separate out aleatoric and epistemic uncertainty is unusual. I was not aware of the Kendall and Gal (NIPS 2017) work, but reviewing that, there are significant differences in how your model is attempting to capture this (notably their use of a Bayesian Neural Network). Therefore, I think you have a higher standard to show that this is doing the right thing.*

In the revised manuscript, we have explained our method of estimated uncertainty in greater detail: Our approach used to estimate the uncertainty follows that of Lakshminarayanan et al (NIPS 2017) in the paper “Simple and Scalable Predictive Uncertainty Estimation using Deep Ensembles”, which was shown to outperform MC Dropout in the original publication. This technique has been widely applied in the broader scientific/medical machine learning literature (e.g. Weigert et al, Nature Methods, 15, 1090 (2018); De Fauw et al., Nature Medicine, 24, 1342 (2018); Tomašev et al., Nature, 572, 116 (2019)).

I do not think the evaluation culminating in Figure 4 accomplishes this. The questions that first come to mind that are left unresolved are:

- a. What is the coverage probability? The figure shows ranking, but that's not what the uncertainty modelled attempted to capture. If the model says the standard deviation of the error is X, is it actually X?*

For regression tasks the model gives an estimate of the standard deviation of its own prediction. Importantly, this predictive uncertainty is not an estimate for the magnitude of the error and so there is no X to compare against.

In some cases it is helpful to introduce a post-hoc calibration factor to attempt to re-scale/calibrate the uncertainty estimates to have the same magnitude as the model error (in which case a coverage could be defined). However, we did not consider it a useful addition to introduce such additional calibration stages as they are not critical for the envisaged materials discovery workflows via active learning. We have added a section in the SI highlighting why the uncertainty estimates do not need to be calibrated for active learning. In this section we also reference existing literature covering different calibration methods and when to use them that readers interested in calibrating their uncertainty estimates could refer to.

b. *What's the relative contributions / magnitudes of the two error components?*

We have introduced the following plot into the SI to explore these ideas.

The first rows of the plot shows the Log of the ratio of the aleatoric and epistemic contributions to the uncertainty. Here we see that for each of the data sets considered one of the two uncertainty components is dominant (if equivalent we would expect symmetric distributions centred on 0). Importantly the dominant source of uncertainty changes depending on the data set. The trade-off between different types of uncertainty depends on whether the test set being considered is in an

interpolative (i.e. typically smaller epistemic uncertainty) or extrapolative regime compared to the training set.

The second row of the plot shows histograms of the different uncertainty components as well as the absolute error. This plot shows clearly that the uncertainty distributions are clearly mis-calibrated when compared to the error-distribution as alluded to in the response to 1.a.

The third row of the plot is a repeat of the second where we have produced curves using a gaussian kernel to allow the shapes of the distributions to be seen more clearly.

c. *Is using one of the other error models insufficient?*

In the model presented here our epistemic uncertainty is not independent of the fact that we are using a NLL based loss function that also provides an estimate of the aleatoric contribution. To address this idea more clearly in the text we have redesigned Fig. 4 (Now Fig. 3).

In the updated figure we compare the between the approach described in the paper that considers both aleatoric and epistemic contributions and a more naive approach (purely epistemic approach) of just taking the uncertainty from the variance of an ensemble trained with a typical L1 loss function. We believe this updated plot is much clearer in the benefits of the full approach as we see the full framework does provide better uncertainty estimates.

2. *The methods section does not describe a hyperparameter search procedure. Depending on the procedure used, it could raise questions about the validity of the results on the test set.*

In the revised manuscript, we have included a fuller discussion on how the model architecture and hyper-parameters were chosen based on cognate models in the machine learning literature. It is important to note that we use the same fixed architecture and hyper-parameters for all the datasets considered in the paper. Therefore, overfitting is highly unlikely. We believe the

reference model described is a good starting point for any end users who may wish to test this approach as an option.

Hyperparameter and neural architecture search for deep attention-based networks is a subject of active research in machine learning [So et al., “The Evolved Transformer”, ICML 2019, arXiv:1901.11117; Tsai et al, “Finding Fast Transformers: One-Shot Neural Architecture Search by Component Composition”, arXiv:2008.06808], and as such we believe this is outside the scope of this manuscript.

Related, while the Roost code is open-sourced, I looked through the repository and did not see a direct way to reproduce the models described in this paper.

I would expect code that directly reproduces the models to be included in some way either in the Roost repository directly or as supplemental files with this publication.

We have now attached a static release of the code as it stands alongside the revised publication. Using this code release the experiments can be directly reproduced by calling the `examples/roost-example.py` script from the top of the repository for each of the sets of command line arguments listed in the `main-experiments.txt` file included in the SI.

The design of the code base was intended to make it as simple as possible to apply the model to problems of an end user's choosing. Open sourcing the code in this way early in its development has resulted in the software already being used by independent research groups. We believe that structuring the codebase independently from this paper will be more helpful in the longer term.

Minor issues

1. *The end of the first paragraph in the introduction: This is not actually fair to how high-throughput workflows actually work. A number of simplifications (like assuming atomic substitutions produce approximately the same crystal structure) are used to avoid the costly global optimization. So it's not a “prohibitive bottleneck”, it's something that is worked around.*

We have amended the text to include reference to prototyping strategies and removed the reference to “prohibitive bottleneck”.

2. *s/statistically principle manner/statistically principled manner/*

Corrected in revised manuscript.

3. *s/n is the initial vector/n is the size of the initial vector/*

Corrected in revised manuscript.

4. *Below equation (2): You refer to $f^t(\dots)$. I suspect this is a typo and should be $f(\dots)$. Unless you are implying a step dependent f function (which is a fine architecture choice), but you need to be clear, including in equation (2)*

We intended for the function to be step dependent and have corrected the notation accordingly.

5. *Associated with equation (2): You really should be clear that these edge terms are directional. e_{ij} is used to update node i . I had to work through the next several equations before that was clear.*

We have added a sentence to explain that the e_{ij} terms are directional based on the concatenation order of h_i and h_j .

6. *Eq (3): Your notation is sloppy, you have overloaded j . The denominator sum should use a different index.*

Corrected in revised manuscript.

7. *Eq (4): It's worth noting in the text that you use a residual net structure. This is a clear design choice that not all graph nets use.*

We have added an explanatory sentence to say that we update the representations in a residual manner.

8. *Eq (5) and Eq (6). You never define F or y_i . This whole part would probably be simpler to use F in equation (5) and avoid $y^{\hat{i}}$*

As part of addressing the first major concern we have re-written this section using a revised notation scheme that we believe is clearer.

9. *Eq (6). $\theta^{\hat{m}}$ should be $\theta^{\hat{w}}$*

Corrected in revised manuscript.

10. *Fig 3: The shading is not defined.*

The shading shows the standard deviations of the 10 runs. We have added a sentence to this effect.

11. *Fig 3: I can't make sense out of the caption. You say it is "error on the test set" but then say "training curve" which is at least ambiguous whether it's training or test set error.*

We have rewritten this section for improved clarity. By "training curve" we were referring to how a variable changes over the course of training the model. In this case the variable in question was the error on the test set. We concede that some resources refer to the plot we call a "training curve" as a "learning curve". However, as we examine the sample efficiency and illustrate it using

the log-log plot in Fig 2. which is also referred to as a “learning curve” calling Fig 3. (Now Fig 4.) a “learning curve” would have overloaded the term.

- 12. The methods section does not contain any details for the transfer learning. In particular, how you either limited fine tuning training (or not) or used frozen layers (or not).*

We allowed the optimizer to freely update all the parameters in the model during our transfer learning experiments. We have updated the manuscript to include this information in the methods section.

REVIEWERS' COMMENTS

Reviewer #2 (Remarks to the Author):

The revisions address all my concerns. I believe the paper is ready for publication.

Reviewer #3 (Remarks to the Author):

The modifications to the paper have filled in important detail and the ablation study has significantly strengthened the empirical evaluation. Given the other reviewers opinion on the wider interest of this work, I have no objections to publication in Nature Communications.

I have a few remaining very minor comments that I hope the authors can address before publication.

* Lines 325-326 specifically refer to "well-calibrated uncertainty estimates" but the text does not acknowledge that your uncertainty estimates are *not* well-calibrated (the analysis in supplementary) which could easily leave the reader with a mistaken impression. The text should acknowledge this and point to supplemental for details.

* Similarly, it's worth acknowledging in the main text the results in supplementary figure 4 (namely that relative magnitudes for aleatoric and epistemic uncertainty vary quite significantly and across datasets tested).

A couple of comments for the authors to think about that probably don't lead to any changes for this paper

* Fig 3: As your text acknowledges, the key values to see in the plot are the derivatives of the curves. Other data visualization would make that more apparent (though computing numerical derivatives here is fraught with so few points).

* Eq 8: I realized after the fact that my comment about coverage probabilities made an assumption that you were estimating a Gaussian (or something other distribution) for the final uncertainty estimate (which Lakshminarayanan et al. do). But you actually aren't making any clear assumption about the form of the final uncertainty. In the absence of a probabilistic assumption, I'm actually not sure that this way of combining variance is correct. This may be worth considering for future work.

Lastly a couple of notes specifically responding to the rebuttal

Hyperparameter and neural architecture search for deep attention-based networks is a subject of active research in machine learning [So et al., "The Evolved Transformer", ICML 2019, arXiv:1901.11117; Tsai et al, "Finding Fast Transformers: One-Shot Neural Architecture Search by Component Composition", arXiv:2008.06808], and as such we believe this is outside the scope of this manuscript.

I want to clarify that I was not suggesting you embark on complex hyperparameter searches. However, it is quite common to brute-force type scan over hyperparameters when doing model building. This can lead to p-hacking like problems where many models are tried (which is why I asked what your procedure was). In your case, your additions to methods make clear that you went with relatively fixed hyperparameters, so you avoid these problems. Thank you for adding the details to methods.

We

have added a section in the SI highlighting why the uncertainty estimates do not need to be calibrated for active learning.

I don't completely agree with you. Yes, the details of the active learning procedure matter tremendously for how much the calibration matters and sometimes you have free parameters that effectively recalibrate this. But very common acquisition functions like Upper Confidence Bound do directly depend on the magnitude of the uncertainty estimates.

We believe that structuring the codebase independently from this paper will be more helpful in the longer term.

Completely agree. Thank you for providing the additional detail.

In the revised manuscript, we have explained our method of estimated uncertainty in greater detail: Our approach used to estimate the uncertainty follows that of Lakshminarayanan et al (NIPS 2017) in the paper "Simple and Scalable Predictive Uncertainty Estimation using Deep Ensembles", which was shown to outperform MC Dropout in the original publication. This technique has been widely applied in the broader scientific/medical machine learning literature (e.g. Weigert et al, Nature Methods, 15, 1090 (2018); De Fauw et al., Nature Medicine, 24, 1342 (2018); Tomášev et al., Nature, 572, 116 (2019)).

Your point about "widely adopted" given your list of citations is misleading here. I reviewed all 3 citations. Deep ensembling as uncertainty estimates is used in all 3 (and I agree ensembling for epistemic uncertainty estimation is common). However, this point was "The attempt to separate out aleatoric and epistemic uncertainty is unusual." Unless I have misread the papers, only Weigert et al. uses aleatoric uncertainty estimates as an output of the model.

Reviewer #3 (Remarks to the Author):

The modifications to the paper have filled in important detail and the ablation study has significantly strengthened the empirical evaluation. Given the other reviewers opinion on the wider interest of this work, I have no objections to publication in Nature Communications.

I have a few remaining very minor comments that I hope the authors can address before publication.

** Lines 325-326 specifically refer to “well-calibrated uncertainty estimates” but the text does not acknowledge that your uncertainty estimates are **not** well-calibrated (the analysis in supplementary) which could easily leave the reader with a mistaken impression. The text should acknowledge this and point to supplemental for details.*

We have removed reference to well-calibrated uncertainty estimates and indicated in the main manuscript that a discussion on calibration is available in the SI.

** Similarly, it's worth acknowledging in the main text the results in supplementary figure 4 (namely that relative magnitudes for aleatoric and epistemic uncertainty vary quite significantly and across datasets tested).*

We have now included a sentence pointing to this variation across data sets in the main manuscript with reference to the figures in the SI.

A couple of comments for the authors to think about that probably don't lead to any changes for this paper:

** Fig 3: As your text acknowledges, the key values to see in the plot are the derivatives of the curves. Other data visualization would make that more apparent (though computing numerical derivatives here is fraught with so few points).*

An area of interest for us is the design of robust metrics that quantify the qualitative ideas shown in the plots. As an area for future work, it could perhaps be easier to define such metrics in the gradient space.

** Eq 8: I realized after the fact that my comment about coverage probabilities made an assumption that you were estimating a Gaussian (or something other distribution) for the final uncertainty estimate (which Lakshminarayanan et al. do). But you actually aren't making any clear assumption about the form of the final uncertainty. In the absence of a probabilistic assumption, I'm actually not sure that this way of combining variance is correct. This may be worth considering for future work.*

We believe the approach taken here is consistent with the law of total variances assuming that the learners within the ensemble are uncorrelated. There may well be choices of architecture where this would not be a reasonable assumption.

Lastly a couple of notes specifically responding to the rebuttal:

Hyperparameter and neural architecture search for deep attention-based networks is a subject of active research in machine learning [So et al., “The Evolved Transformer”, ICML 2019, arXiv:1901.11117; Tsai et al, “Finding Fast Transformers: One-Shot Neural Architecture Search by Component Composition”, arXiv:2008.06808], and as such we believe this is outside the scope of this manuscript.

I want to clarify that I was not suggesting you embark on complex hyperparameter searches. However, it is quite common to brute-force type scan over hyperparameters when doing model building. This can lead to p-hacking like problems where many models are tried (which is why I asked what your procedure was). In your case, your additions to methods make clear that you went with relatively fixed hyperparameters, so you avoid these problems. Thank you for adding the details to methods.

We have added a section in the SI highlighting why the uncertainty estimates do not need to be calibrated for active learning.

I don't completely agree with you. Yes, the details of the active learning procedure matter tremendously for how much the calibration matters and sometimes you have free parameters that effectively recalibrate this. But very common acquisition functions like Upper Confidence Bound do directly depend on the magnitude of the uncertainty estimates.

We believe that structuring the codebase independently from this paper will be more helpful in the longer term.

Completely agree. Thank you for providing the additional detail.

In the revised manuscript, we have explained our method of estimated uncertainty in greater detail: Our approach used to estimate the uncertainty follows that of Lakshminarayanan et al (NIPS 2017) in the paper “Simple and Scalable Predictive Uncertainty Estimation using Deep Ensembles”, which was shown to outperform MC Dropout in the original publication. This technique has been widely applied in the broader scientific/medical machine learning literature (e.g. Weigert et al, Nature Methods, 15, 1090 (2018); De Fauw et al., Nature Medicine, 24, 1342 (2018); Tomašev et al., Nature, 572, 116 (2019)).

Your point about “widely adopted” given your list of citations is misleading here. I reviewed all 3 citations. Deep ensembling as uncertainty estimates is used in all 3 (and I agree ensembling for epistemic uncertainty estimation is common). However, this point was “The attempt to separate out aleatoric and epistemic uncertainty is unusual.” Unless I have misread the papers, only Weigert et al. uses aleatoric uncertainty estimates as an output of the model.

In Lakshminarayanan et al (NeurIPS 2017) they make 3 suggestions: “We suggest a simple recipe: (1) use a proper scoring rule as the training criterion, (2) use adversarial training [13] to smooth the predictive distributions, and (3) train an ensemble”. The adversarial training aspect (2) included by the original authors is often omitted, as we did, due to the limited benefit it’s shown to have in the original paper.

Therefore, the key distinction between the “Deep Ensemble” approach and ensembling is the use of a proper scoring rule (1). For classification tasks the cross-entropy loss is a proper scoring rule hence most ensemble models for classification are “Deep Ensembles” in the spirit of Lakshminarayanan et al. (NeurIPS 2017) - all references cite the original work. However, the MSE/MAE are not proper scoring rules for regression tasks. On the other hand, the so-called robust losses that incorporate a predicted aleatoric uncertainty are proper scoring rules for regression tasks. Of the included references only Weigert et al. examines a regression task which is why it is the only one of the three that explicitly considers aleatoric effects. We have added a statement and citation explaining that deep ensembles require proper scoring rules and that the losses incorporating the aleatoric uncertainty are proper scoring rules.